# Plasma FIB milling for the determination of structures in situ

Casper Berger [1,2,5], Maud Dumoux [1,5], Thomas Glen[1,5], Neville B.-y. Yee[3], John M. Mitchels[4], Zuzana Patáková [4], Michele C. Darrow[3], James H. Naismith[1,2] & Michael Grange [1,2] ✉

Structural biology studies inside cells and tissues require methods to thin vitrified specimens to electron transparency. Until now, focused ion beams based on gallium have been used. However, ion implantation, changes to surface chemistry and an inability to access high currents limit gallium application. Here, we show that plasma-coupled ion sources can produce cryogenic lamellae of vitrified human cells in a robust and automated manner, with quality sufficient for pseudo-atomic structure determination. Lamellae were produced in a prototype microscope equipped for long cryogenic run times (> 1 week) and with multi-specimen support fully compatible with modern-day transmission electron microscopes. We demonstrate that plasma ion sources can be used for structural biology within cells, determining a structure in situ to 4.9 Å, and characterise the resolution dependence on particle distance from the lamella edge. We describe a workflow upon which different plasmas can be examined to further streamline lamella fabrication.

Cryo-electron tomography (cryoET) enables the structural study of macromolecular complexes within their native cellular environment. Understanding the structural landscape of macromolecules in relation to their subcellular environment is key in linking structure to function. Since most biological samples are too thick to directly image with a transmission electron microscope (TEM), methods to thin down biological material to the <300 nm necessary for electron transparency have been instrumental for enabling cryoET of the internal regions of cells. Focused ion beam (FIB) milling was originally developed for material science applications, such as microchip manufacture, at room temperature, where its ability to precisely shape objects on the nanoscale revolutionised electronics[1,2]. More recently, the approach has been adapted for use with biological samples through the adaptation of existing dual beam FIB/scanning electron microscopes (FIB/SEM) with cryogenic stages enabling frozen-hydrated samples to be shaped and thinned to thicknesses enabling subsequent cryoET analyses[3,4]. This capability has enabled intracellular structural studies on a wide variety of biological samples to pseudo-atomic resolution[5–9].

Current FIB/SEM instruments used to prepare FIB lamella are not optimised for cryogenic applications[10–12]. The large sample chamber (~40 L), originally designed for 8" Si wafers at room temperature, has a vacuum typically of the order $10^{-6}$ mbar at ambient temperature, dropping by a ~10-fold under cryogenic (~100 K) conditions. This results in ice (re)deposition, with rates of up to 85 nm/h reported[11,13]. This limits the number of lamellae that can be prepared on one grid in an experiment and impacts the proportion of successful lamella[11]. Studies have recently demonstrated success with automated FIB lamella fabrication[11,12,14,15], but use of gas cooled systems limits length of overnight runs and ice growth effectively limits the number of lamellae that can be prepared in one automated session.

Lamella fabrication to-date for structural biology has typically been performed with a gallium FIB. Gallium works well as a metal ion source due its low melting point, low volatility, low vapour pressure, good emission characteristics and good vacuum properties[16]. Gallium also sputters a range of different materials efficiently. The point source nature of a gallium ion source enables the beam to be tightly focussed[17]

[1]Structural Biology, The Rosalind Franklin Institute, Harwell Science & Innovation Campus, Didcot OX11 0QS, United Kingdom. [2]Division of Structural Biology, Wellcome Centre for Human Genetics, University of Oxford, OX3 7BN Oxford, United Kingdom. [3]Artificial Intelligence & Informatics, The Rosalind Franklin Institute, Harwell Science & Innovation Campus, Didcot OX11 0QS, United Kingdom. [4]Thermo Fisher Scientific Brno s.r.o, Brno 627 00, Czech Republic. [5]These authors contributed equally: Casper Berger, Maud Dumoux, Thomas Glen. ✉e-mail: michael.grange@rfi.ac.uk

enabling control at the tens to hundreds of nanometer scale. However, at higher beam currents the beam becomes increasingly divergent, limiting the bulk milling these beams are capable of. Implantation is likely from any focussed ion beam[18], but gallium has been shown to react with various samples, migrate and significantly alter the original structure[18–20]. Alternative approaches are therefore of interest to reduce these effects.

One such alternative is inductively coupled plasmas generated from a gas, e.g., nitrogen, oxygen, xenon, or argon[21]. Under low beam current regimes, plasma sources have comparable probe sizes to gallium but under high beam current regimes plasma species have smaller probe sizes than gallium[22]. This means that in principle plasma beams are highly suited to milling large volumes at high currents but remain capable of the fine milling required for thinner, more fragile, lamella used in life sciences[23].

Here, we describe a protocol for automated plasma focused ion beam (PFIB) milling lamellae of cryogenic cellular samples. We were able to hold multiple specimens at cryogenic temperatures at low contamination rates (<2 nm/hr) for weeks. We characterised the milling rates of plasma ion sources (O, N, Ar, Xe) at a range of currents. We demonstrate that an argon plasma source can produce lamellae with a high success rate (85%). The lamellae were used to generate a cryoET dataset of hundreds of tomograms, which show thicknesses, features, and resolution on a sub-nanometre scale, enabling sub volume averaging of the human 80 S ribosome to a resolution of 4.9 Å, with the well-ordered regions of the structure at resolutions close to the Nyquist limit of 3.8 Å. We further show that there is a dependence on the attainable resolution on the distance of particles from the top and bottom PFIB milling surfaces. Our study demonstrates that a plasma FIB/SEM can enable in situ structural biology at pseudo-atomic resolution ranges and paves a way for higher throughput.

## Results

### A plasma FIB/SEM for multi-sample imaging over long time scales

We used a custom designed plasma FIB/SEM microscope (Supplementary Fig. 1) equipped with a coincident PFIB and SEM inside a chamber with redeposition rates <2 nm/hr (Supplementary Fig. 2), and a stage with rotational freedom of +14° to −190°. The sample chamber vacuum volume is ~6 litres and is maintained at a pressure of ~1 ×10$^{-7}$ mbar at cryogenic temperature. Up to 12 clipped grids can be mounted into a multi-specimen cassette which can be robotically loaded into the chamber, one at a time (Supplementary Fig. 1). The PFIB can be configured to be used with different ion sources, including xenon, oxygen and argon. The cryo box (anti-contaminator) and stage are braid cooled from liquid nitrogen dewars that are filled automatically when needed, allowing continuous cryo runtimes of up to a week or more. The characteristics of this microscope, including minimal ice growth (Supplementary Fig. 2), robotic sample transfers and long cryo runtimes enabled continuous preparation of PFIB lamellae on biological specimens for up to 1 week. In effect the number of lamellae that can be produced is sample, rather than machine dependent. The high-vacuum exchange of samples and robotic sample entry reduce the risk for sample damage or disruption from handling.

### Milling rates of plasma ions on vitreous cellular samples

Determining the milling rate (or sputter yield) of each plasma beam can inform the design of milling protocols. Others have already characterised the milling rates for different materials[24], but not on vitrified biological samples. We used a second microscope (Thermo Scientific Helios Hydra with gas-cooled cryogenic stage) with the same ion column as the first to determine milling rates more accurately for xenon, nitrogen, oxygen, and argon, at normal incident angles. Based on 3 experimental repeats on different regions of the same plunge-frozen yeast sample, our measurements show that for vitrified biological

samples xenon has the highest milling rate of $16.8 \pm 0.2$ μm³/nC. Nitrogen and oxygen have values of $10.6 \pm 0.2$ and $10.0 \pm 0.4$ μm³/nC respectively, while argon is $4.3 \pm 0.21$ μm³/nC (Supplementary Fig. 3; Table 1). This compares with a milling rate of 7.7 μm³/nC for gallium in ice[25]. These milling rates are ~20–100 times higher than those reported at 30 kV in silicon for these plasma beams[24]. This is of the same order as the factor of ten that is suggested within the community as a rule of thumb for the difference between milling in vitrified ice and silicon[26]. During lamella fabrication for cryoET, typical FIB milling angles can range between 8–13°, with angles up to ~25° being used for some applications. We measured the milling rate of vitrified samples using xenon and argon at currents between 20 pA up to 2 nA (currents typically used for lamella fabrication[13,27]) at angles from 10° to 40°, and calculated milling rates. (Supplementary Fig. 3; Table 2). Our results show that at high angles, there was a greater difference in milling rates between the two gases (at 90° xenon is 3.7x greater) compared to shallow angles (10° xenon is 2.7x greater), suggesting that the incidence angle influences milling efficiency in vitrified water, and that this dependence varies with milling gas.

### Plasma ion source for automated, high throughput lamella fabrication

While xenon has the greater milling rate, we opted to use argon for our first experiments of fully automated lamella fabrication. This is because we had accumulated more experience in the control of argon plasma to produce smooth flat surfaces for cryo-serial plasma FIB/SEM[28]. We reasoned that smoother lamella surfaces would correlate with better quality tomography data and therefore a better initial test of the approach. Since the instrument is not limited by ice redeposition in the chamber for these initial experiments the additional time taken by argon was not critical. To determine how argon performs in automated lamella fabrication, we prepared three independent datasets of lamellae of plunge-frozen (Chlamydia-infected) HeLa cells using automated lamella fabrication, and determined the success rates and milling times, (Fig. 1; Supplementary Fig. 4 and 5). The average time to prepare a lamella was ~45 mins; 32 min of coarse milling and 13 min of

**Table 1 | Measured miling rates taken from triplicate measurements on plunge-frozen yeast samples**

| Plasma | Xenon (MW 131.29) | Nitrogen (MW 14) | Oxygen (MW 15.99) | Argon (MW 39.95) |
|---|---|---|---|---|
| Milling rate (μm³/nC) | 16.7 ± 0.2 | 10.6 ± 0.2 | 10.0 ± 0.4 | 4.3 ± 0.1 |

Milling rates for xenon, nitrogen, oxygen and argon are shown. Milling rates were measured as described in Supplementary Fig. 3. Errors shown are the standard error derived using the least squares method from a linear line of best fit that passes through the origin. Source data are provided as a Source Data file.

**Table 2 | Measured milling rates taken from triplicate measurements on plunge-frozen HeLa samples with varying milling angles**

| Milling angle (°) | Xenon (MW 131.29) Milling rate (μm³/nC) | Argon (MW 39.95) Milling rate (μm³/nC) |
|---|---|---|
| 90 | 16.7 ± 1.3 | 4.5 ± 0.3 |
| 40 | 34.1 ± 3.0 | 14.4 ± 1.4 |
| 30 | 52.4 ± 7.9 | 23.1 ± 0.7 |
| 20 | 80.1 ± 10.4 | 33.3 ± 2.7 |
| 10 | 157.6 ± 16.1 | 59.3 ± 12.1 |

Milling rates for xenon and argon are shown. Milling rates were measured as described in Supplementary Fig. 3. Errors shown are the standard error derived using the least squares method from a linear line of best fit that passes through the origin. Source data are provided as a Source Data file.

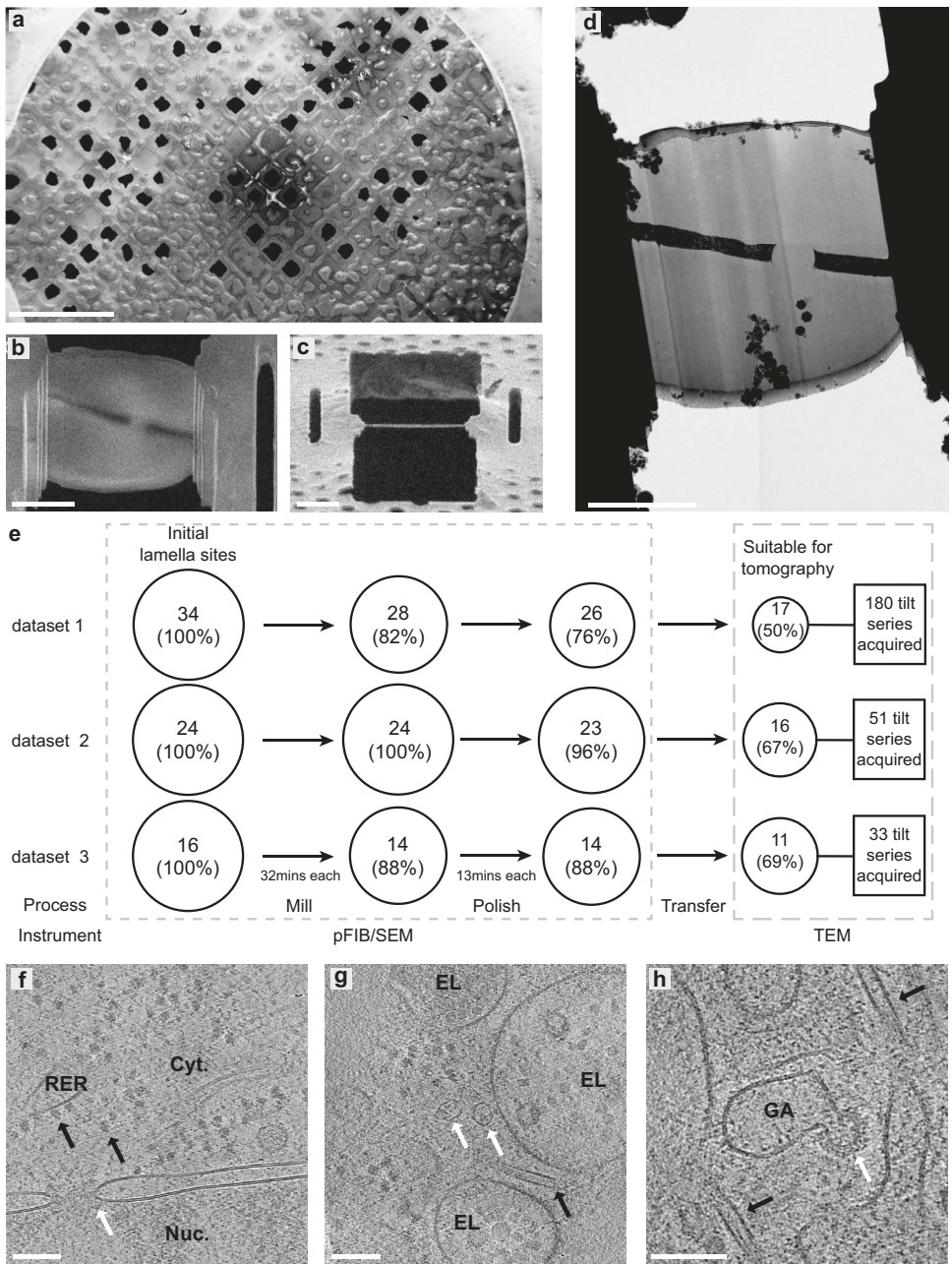

**Fig. 1 | Overview of lamella production using plasma FIB milling. a** SEM overview of a representative uncoated grid (cells were grown on UltraAuFoil (Quantifoil) grids) from three independent plasma FIB milling sessions. **b** Representative SEM image of a finished lamella and **c** its respective PFIB image. **d** Low magnification TEM image of the same lamella. Scale bar are 500 μm in **a** and 5 μm in **b**, **c** and **d**. The chart in (**e**) shows the number of successful lamella sites at each stage for three independent datasets with the time taken between coarse and finer milling steps. The cumulative success rate for all 74 initial lamella sites are: course milling (89%), polishing (85%), suitable for tomography (59%). Source data are provided as a Source Data file. **f**–**h** Representative tomographic slices containing several sub-cellular features, including (**f**) nucleus (Nuc.) and cytosol (Cyt.) with a nuclear pore (white arrow) in the nuclear envelope. Ribosomes (black arrows) are visible both free within the cytosol and tethered to the rough-endoplasmic reticulum (RER). scalebar: 100 nm; **g** endo- lysosomal vesicles (EL), a microtubule (black arrow) and vault complexes (white arrows). Scale bar: 100 nm; **h** microtubules (black arrows) and vesicle budding from the Golgi (GA) via a retromer coat (white arrow). The full tomographic volume for this slice is shown in Supplementary Video 1. Scale bar: 100 nm.

fine milling, of which ~25 and 10 min, respectively, represented steps where material was milled (details in Table 3).

Once completed the grids were transferred to a compatible TEM in the same cassette, preserving the milling angle close to perpendicular to the tilt axis (robotic automation still leads to some slight rotational variability upon loading), eliminating need for manual handling. No lamellae were lost during sample transfer between PFIB and TEM.

From 3 independent milling sessions we targeted 74 sites for automated lamella fabrication. Of the 74 total lamella sites, 63 (85%) produced intact lamella after the final polishing step based on the SEM images. The target thickness determined empirically was ~200 nm (see methods for details). Of these, 44 (59%) were considered suitable for performing tomography (excluding lamella with incomplete vitrification, cracks, or unsuitable acquisition areas (e.g nucleus)). A total of 180 tilt-series were acquired on the first dataset for subsequent sub

**Table 3 | Milling parameters used for lamella preparation**

| Step | Milling current (pA) | Width (µm) | Height (µm) / Overlap (%) | Offset from lamella (nm) | Milling depth (µm) | Drift correction interval (s) |
|---|---|---|---|---|---|---|
| Rough milling | 2000 | 15 (top) 14 (bottom) | 7 (top) 7 (bottom) | 1500 | 1.875 | 500 |
| Medium milling | 200 | 13.3 (top) 13 (bottom) | 200% | 600 | 1.41 | 500 |
| Fine milling | 60 | 12.7 (top) 12.2 (bottom) | 200% | 300 | 1.125 | 60 |
| Polish 1 | 60 | 12 | 200% | 150 | 0.5625 | 30 |
| Polish 2 | 20 | 12 | 200% | 0 | 0.5625 | 10 |

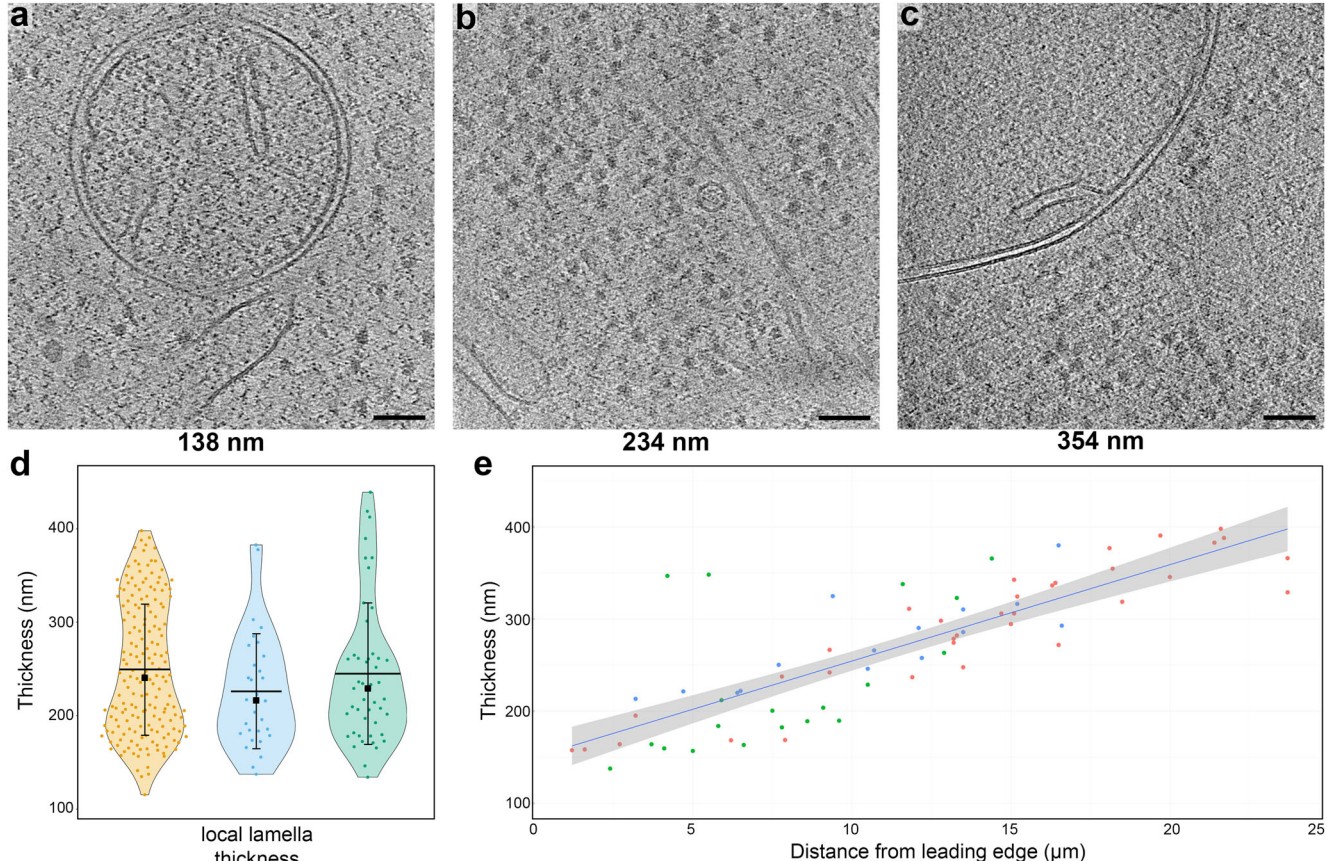

**Fig. 2 | Plasma-milling produces lamellae with a range of thicknesses that exhibit usable sub-cellular tomographic data. a–c** representative tomographic slices of tomograms recorded on lamellae with different thicknesses (indicated below each panel). Scale bars: 100 nm. **d** Violin plots showing the distribution of tomogram thickness as determined in the reconstructed tomograms, in three independent argon datasets prepared with the same milling conditions. Orange (left): mean SD; 250 nm ± 70, $n = 180$ tomograms (from 17 lamellae). Blue (middle): mean SD; 227 nm ± 62, $n = 30$ tomograms (from 11 lamellae). Green (right): mean SD; 245 nm ± 76, $n = 47$ tomograms (from 16 lamellae). Horizontal black bars indicate the mean thickness, error bars the standard deviation and the black square the median (from left to right: 241 nm, 217 nm, 229 nm). Source data are provided as a Source Data file. **e** Scatter plot for tomogram thickness and the distance to the front of the lamellae for all tomograms recorded on three different lamellae from the first dataset (datapoints coloured in green, red and blue for each lamella). A linear trend line (blue) is shown, with the 0.95 confidence interval for the trend line shown in grey. Plots for eight individual tomograms are also shown in Supplementary Fig. 6a, and for all the tomograms from the other two datasets in Supplementary Fig. 6b, c. Source data are provided as a Source Data file.

volume averaging, and 51 and 33 tilt-series on dataset 2 and 3, respectively, were used for assessing the lamella quality and thickness (Fig. 2).

**Electron cryo-tomography of plasma FIB milled lamellae**
Tomograms acquired from the lamellae generally exhibited characteristics associated with vitrified cellular tomograms (no Bragg reflections within images, no aggregation-segregation induced contrast, uniform rounded membranes). We could observed a wide variety of macromolecular complexes (Fig. 1f–h, Supplementary Video 1) in the reconstructed tomograms including nuclear pore complexes[29,30],

microtubules[31], vault complexes[32], and vesicle budding via retromer coat proteins[33]. The reconstructed tomograms had an overall mean thickness of 247 nm ± 71 nm for the recorded tilt-series on all three datasets (Fig. 2). The lamellae generated here were suitable for successful TEM tilt-series acquisition, observing cellular features in tomograms >350 nm thick (Fig. 2).

We measured the thickness of all lamellae and the distance from the leading edge at which the tomogram was acquired (Fig. 2; Supplementary Fig. 6). Interestingly, most lamellae showed a thickness gradient[13], with some lamella being twice as thick at the back of the lamella as toward the front (<200 nm vs >350 nm), while others were

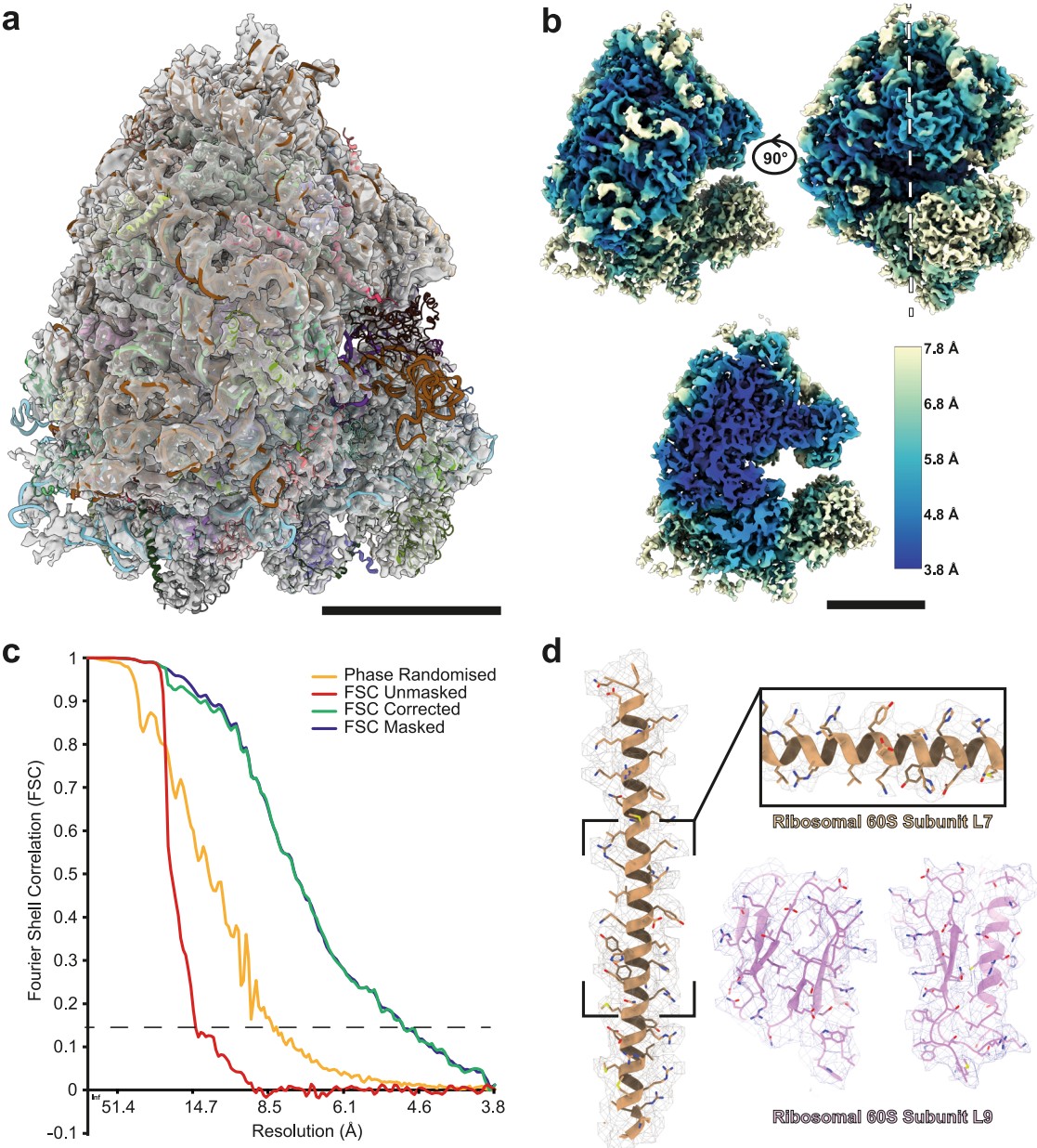

**Fig. 3 | Structure of the human 80 S ribosome obtained from cellular tomograms. a** Density map of the 80 S ribosome with fitted single-particle determined model of the human ribosome obtained from isolated ribosomes (PDB: 4UG0[53]) scale bar: 10 nm. **b** The same density map colour-coded for the local resolution from the outside (top) and a central slice (bottom, white line) (scale bar: 10 nm). **c** Fourier shell correlation for the 80 S ribosome gives a masked global resolution of 4.9 Å (dashed line, FSC 0.143). **d** Representative fits into the EM density for different ribosomal regions (60 S subunits L7 (gold) and L9 (magenta)), suggesting quality of the map.

relatively flat. The lamella with the greatest difference in thickness were the longest (Supplementary Fig. 6). There is a need to optimise the protocol to best reflect the cellular size/morphology to achieve uniformly flat lamella. In these experiments, no over tilt was used in coarse or fine milling steps, which has been shown to improve lamella flatness[13].

**Sub-volume averaging at sub-nm and PFIB milling surface effects**

We determined structures of the human ribosome within HeLa cells, using the 180 tilt-series acquired on the first argon datasets. We could obtain a structure of the human 80 S ribosome using 15628 particles from 17 lamellae with a global resolution of 4.9 Å, and local resolution of 3.8 Å for large areas of the 60 S subunit (Fig. 3; Supplementary Fig. 7;

Supplementary Movie 2 and 3). During refinement of the structure to sub-nm resolutions, the 60 S and 40 S diverged in resolution, with the 60 S approaching a resolution close to the Nyquist information limit (3.8 Å), while the 40 S was a resolution above 8 Å. As the particle population is taken from non-arrested HeLa cells, our data captures the native conformational states of the ribosome during translation. These data are available on EMPIAR (EMPIAR-11306) should a future study wish to probe human ribosome dynamics further.

Ion beams are known to cause damage to different materials at the milling surface with a depth of tens of nanometers[34,35], but how ion beams affect the surfaces of biological samples is currently unknown. We therefore interrogated the effect of plasma on damage propensity close to the top and bottom milling surfaces of the lamella. It is difficult to experimentally determine the damage

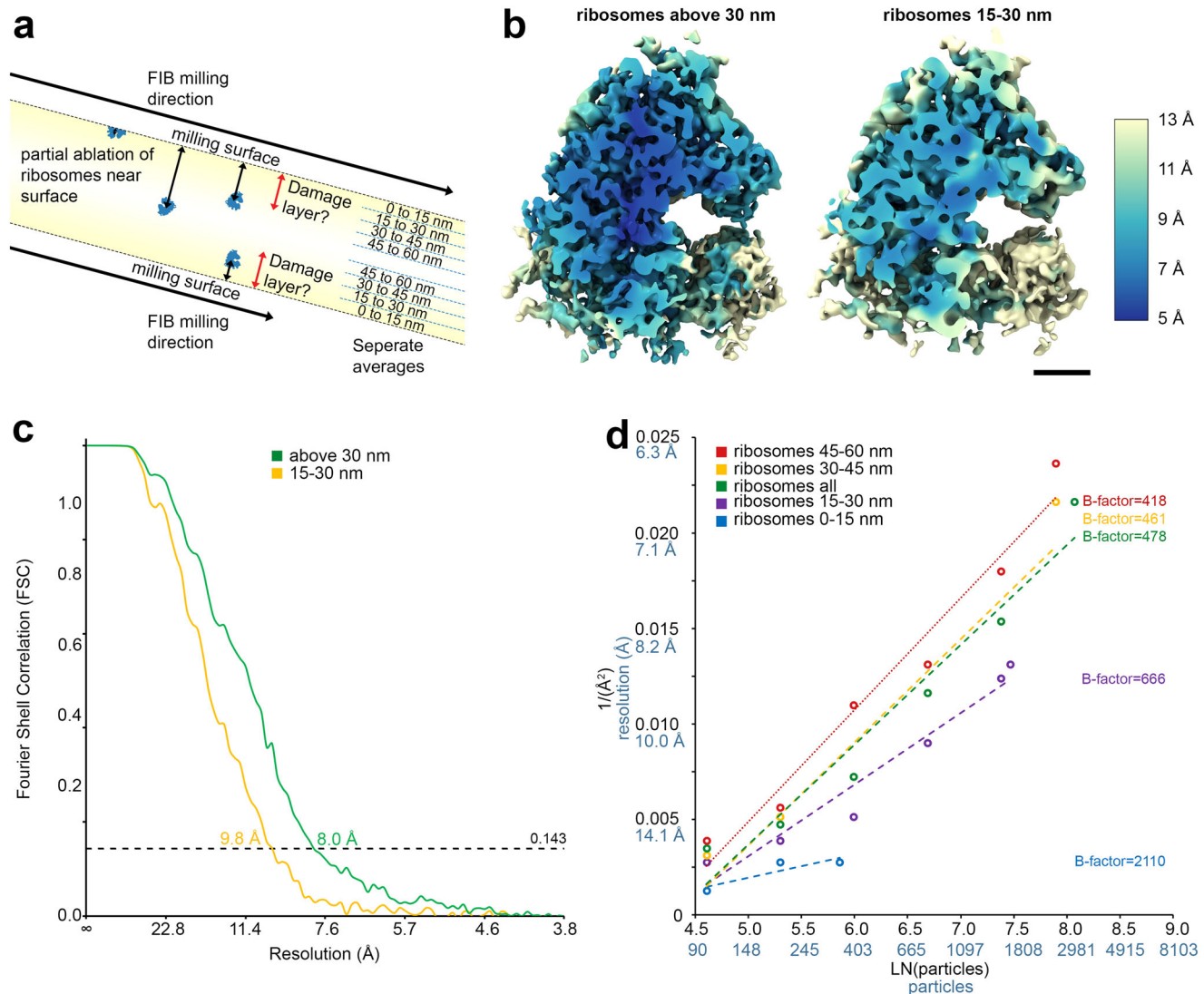

**Fig. 4 | Effect of distance from the lamella edge on the ability to determine ribosome structures. a** Schematic overview of ribosomes in a tomogram of a lamella and how the shortest distances of ribosomes to the top or bottom milling surfaces were obtained. These distances were used to create separate sub volume average ribosome structures in 15 nm increments away from the milling surfaces, to assess the extent and depth of the damage layer caused by PFIB milling. **b** Ribosome structures were determined from acquired tomograms, where 1748 particles taken from >30 nm away from the PFIB milling surfaces (left) and between 15 and 30 nm away from the PFIB milling surfaces (right). Ribosomes are coloured by the local resolution of the maps. Scale bars are 5 nm. **c** FSC curves are shown for the two maps, with a global resolution of 9.8 Å for the ribosomes within 15–30 nm of the edge (orange) and 8 Å for ribosomes outside of this distance (green). **d** B-factors determined for all ribosomes (green) 45–60 nm (red), 30–45 nm (orange), 15 to 30 nm (purple) and 0 to 15 nm (blue) distance to the lamella edge, after sub-volume averaging. These are calculated as 2 over the slope ($Å^2$). Source data are provided as a Source Data file.

induced by an ion beam during lamella fabrication for life science samples; diffraction methods are not possible on cellular samples and sub-volume averaging has, until recently[8,36], been limited to low resolution. The ability to produce structures at sub-nm resolutions offers the opportunity to infer damage done by ion beam milling by aligning particles according to depth and determining the resolution of the alignment. Boundary models for all 180 tomograms were produced and then interpolated to form a geometric model of the tomograms. This enabled robust modelling of the lamella geometry local to each tomogram and to determine the distance of each particle to the nearest milling boundary (Supplementary Fig. 8). When we compared the number of particles after 3D classification with the number after picking, there was an appreciable loss of particles within 20 nm from the lamella edge (Supplementary Fig. 9). The particle positions taken from sub-volume averaging of 3D classified 15,628 ribosomes across 180 tomograms which were then aligned, and sub-volume averaged based on their distance to the nearest

milling boundary for 0-15 nm, 15-30 nm, 30-45 nm and 45-60 (Fig. 4a; Table 4). As controls, the same number of particles above the upper limit of each range (e.g. >15 nm, >30 nm etc.), matched to be from the same tomograms were also aligned and sub-volume averaged. Each set of particles were then independently subjected to sub-volume averaging using a new global 3D refinement. We observed lower resolutions for particles closer to the milling surfaces compared to the matched control particles further away for 0 to 15 nm (21.4 and 13.2 Å respectively with 351 particles) and 15 to 30 nm (9.8 and 8.0 Å respectively with 1748 particles) (Fig. 4b, c; Supplementary Fig. 10; Table 4). To further indicate the extent to which the ion beam impacts particle quality we determined B-factors[37] for each reconstruction in the 15 nm shells and the matched controls (Fig. 4d; Supplementary Fig. 11). These findings show that particles closer to the lamella milling surfaces, up to a depth of 30 to 45 nm, contain less information compared to particles located more centrally. This effect becomes more impactful on particle quality as particles are found

**Table 4 | Effect of particle to milling surface distance on sub-volume averaging resolution**

| Distance Range | Ribosome number | Resolution (Å) | B-factor | Weighted average lamella thickness (nm) |
|---|---|---|---|---|
| 0–15 nm | 351 | 21.4 | 2110 | 193 ± 39.1 |
| >15 nm matched control | 351 | 13.2 | 840 | 193 ± 38.9 |
| 15–30 nm | 1748 | 9.8 | 666 | 190 ± 39.1 |
| >30 nm matched control | 1748 | 8.0 | 441 | 190 ± 39.1 |
| 30–45 nm | 2679 | 7.6 | 461 | 195 ± 42.7 |
| >45 nm matched control | 2679 | 7.3 | 427 | 195 ± 42.7 |
| 45–60 nm | 2951 | 7.3 | 418 | 197 ± 42.7 |
| >60 nm matched control | 2951 | 7.6 | 457 | 207 ± 43.5 |
| All particles | 3200[a] | 7.6 | 478 | 212 ± 48.2 |

Ribosomes were separately sub-volume averaged and the B-factors were determined, based on the nearest distance to one of the PFIB-milled surfaces. Columns list the distance range of the ribosomes, number of 80 S ribosomes used for sub-volume averaging, the obtained resolution, the determined B-factor and the weighted average lamella thickness, which is the average thickness of the tomogram for each particle used in each sub volume average. Source data are provided as a Source Data file.
[a]Maximum number of particles used to determine the B-factor.

closer to the milling surfaces. The highest quality (undamaged) particles reside beyond 45 nm from the milling surfaces.

## Discussion

The determined milling rates for each plasma (Table 2; Supplementary Fig. 3) are non-linear with molecular weight (MW) of the used gas. While xenon (MW 131.29) has the greatest milling rate and largest MW, argon (MW 39.95) has the lowest milling rate while weighing more than nitrogen (MW 14) and oxygen (MW 15.99). The milling rates for oxygen and nitrogen are the same within the error of our measurements. While there is an expectation that a greater mass for ion species should correlate with greater milling rate, the milling rate for different plasmas has been demonstrated to be material dependent[24]. Our analysis of milling rates on plunge-frozen biological samples shows that closer to the glancing angle (smaller milling angles) for argon and xenon the milling rates increase greatly. The generally accepted theory for this is that shallower collision cascades create a greater density of displaced atoms that can potentially be sputtered (as they are closer to the surface), leading to an increase in milling rate at more grazing incident angles[38]. However, approaching very grazing incident angles (ie. close to 0° milling angle) the number of ions penetrating the surface drops rapidly, as more are reflected from the surface. Interestingly, the angular dependence of the milling beam relative to the sample is more pronounced for argon than for xenon (13-fold increase for argon vs 9.4-fold increase for xenon from 90° to 10°). Our observations are consistent with published simulations[39] but mean that, in terms of milling rate, at very shallow angles the choice of gas becomes less critical (between Xe and Ar). Other contributory factors, such as curtaining propensity (either based on probe characteristic or defocus variation) should therefore be considered to a greater extent at shallow angles.

An open question is whether PFIB can be used autonomously to fabricate lamella suitable for in situ structural biology. We report a protocol that produces lamella with a mean thickness of 247 ± 71 nm that runs unsupervised, and show that the lamella produced in this way yield information-rich tomograms. Although manually prepared FIB lamellae can be thinner[40], fully automated lamella prepared with gallium have been reported to have a similar thickness to those reported in the current study[12]. The automated milling protocol used in this study was designed for robust milling on eukaryotic cells, which can vary considerably in thickness, with a high success rate using currents similar to those typically used for lamella fabrication with gallium. Although we focus in this study on using argon for lamella preparation, it's likely that with further optimisation of automated milling procedures similar results could be obtained with xenon, but with potentially higher milling rates than for argon. At relevant milling angles (10–20°) (Fig. S3) milling times for xenon would be a factor of 2.6x

quicker than those for argon. Sample-specific optimisation (for depth) of milling times and careful evaluation of the use of higher currents for milling may also yield faster and/or thinner lamellae fabrication. However, the choice of ion species and/or higher currents and their effects on lamella and tomography data quality will be an important consideration. Further optimisation of the software that controls the automated milling to minimise time not spent milling will also be important in increasing the rate of lamella production.

The mean free path of an electron at 300 kV in water is ~300 nm[41,42]. This limits the thickness of samples that can be imaged by transmitted electrons. Traditionally lamella below 300 nm are prepared to achieve sufficient contrast for cryoET; a simple rule of thumb is that the thinner the lamella the better the contrast. Although higher in contrast, such thin lamellae (<150 nm) have a greater proportion of the volume subject to milling induced effects and images a smaller volume of biological space reducing contextual information. Based on our results with respect to the impact of the ion beam on particle damage close to the milling surface (Fig. 4), thinner lamella will result in a greater proportion of the lamella containing particles of lower quality for sub-volume averaging; this trade-off will be important when designing milling protocols where intermediate thicknesses may be advantageous. Nonetheless, if required, thinner lamellae may be achieved with manual polishing after automated coarse milling.

The very high vacuum in both the plasma FIB/SEM and the TEM instrument used here have practically eliminated redeposited ice contamination in the chamber. Thus, the milled lamella thickness that we fabricate is virtually the same thickness of the lamella that is measured during the TEM experiment. This is in contrast to other instruments where the lamella measured by TEM are often the milled thickness plus a layer of (contrast reducing) redeposited material[11]. The transfer between the autoloading devices on the machines employs a portable cassette filled with liquid nitrogen at atmospheric pressure and some ice contamination was often seen on some of our early lamella (Fig. 1). Subsequently using careful precautions, (low humidity, heating and drying) of all equipment and freshly (<2 hrs) dispensed liquid nitrogen we have been able to greatly reduce ice contamination (supplementary Figs. 4 and 5). Vacuum transfer between microscopes would significantly simplify the process and largely eliminate transfer ice contamination.

From 180 tomograms recorded on the first argon dataset it was possible to use sub tomographic averaging to obtain a structure of the human ribosome to 4.9 Å within lamellae. This compares extremely well with the current contemporary studies in cryo-tomography; ribosomes in isolated bacteria to 3.5 Å[36], ribosomes from lamella of *Saccharomyces Cerevisiae* (Yeast) to a resolution of 5.1 Å[43] and recently lamella of isolated vertebrate myofibrils where the thin filament was determined to 4.5 Å resolution[8]. For the dataset used for sub volume

averaging, the milling process was complete in 2 days and the TEM data acquisition in 4 days. Thus, at a practical level, the protocol opens a route to rapid in situ structural biology with resolutions at or exceeding the highest currently available.

A robust determination on the effect of FIB milling on the damage caused to the surface of lamellae (structural changes, e.g. amorphisation if a crystal lattice, rather than devitrification) is yet to be reported. Ribosomes populate much of the cell making them an attractive model to assess the effect of milling. We were able to show via sub-volume averaging, that particles near the lamella milling surfaces contain less high-spatial frequency information compared to particles towards the middle of the lamellae. In the 15 to 30 nm closest to the top or bottom surface of the lamella, averaging the ribosomes yields the resolution of 9.8 Å vs 8.0 Å for those particles found in the middle of the lamella, matched to the same lamella where possible to exclude effects of local lamella thickness and quality. This was also evident from the calculated B-factors for reconstructions of ribosomes from bands of distances to the PFIB milling surfaces (Fig. 4). Taken together, these data suggest that argon plasma, even at low currents, leads to some structural change within the first 30 to 45 nm of the milling surfaces, with less damage towards the middle of the lamellae. Since the diameter of the ribosome is ~25 Å, the lower resolution observed in the first 15 nm from the milling surfaces is likely to be a combination of partial ablation of ribosomes and damage from argon ions. These results are empirical evidence for plasma beam damage of biological samples. This analysis also establishes a standard that can be used by us and others to robustly evaluate other milling protocols, other plasma chemistries and gallium, which remain to be characterised.

Based on findings from material science applications[34,35], gallium is also likely to create a damage layer near the milling surfaces for vitreous biological samples, quite possibly with a greater depth. The choice of ion species may also impact data quality through implantation. Implantation of the milling ion into lamellae during fabrication will increase the proportion of inelastic vs elastic scattering when imaged in the TEM, reducing image contrast. Argon has over 1/3 less electrons than gallium thus for an equivalent level of implantation argon would be expected to be intrinsically less distorting due to lower local change in potential derived from the implanted ion species. Accurate measurement of gallium and argon implantation upon vitrified samples requires scanning electron methods, atom probe tomography and/or electron energy loss spectrometry that are beyond the scope of this work. Notably both these approaches require very large electron fluences (2000–3000 $e^-/Å^2$) which would severely damage the sample.

Electron cryo-tomography is the frontier of structural biology but there is a pressing need to streamline workflows to enable its widespread adoption. Here we present a robust method for lamella fabrication using plasma FIB. This demonstrates the use of plasma for usage in lamella fabrication for cryoET. Our results show that a plasma cryo FIB/SEM with low contamination rates and with a robotic multi-specimen loading device can streamline the fabrication of lamellae for pseudo-atomic structure determination. In our first attempt using argon plasma, we produced around 20 high quality lamella per day. We have accurately quantified a relationship between particle distance from the lamella milling surfaces and achievable resolution via sub volume averaging, implicating that damage from plasma milling with argon has a depth of 30 to 45 nm. We conclude that plasma FIB lamella fabrication is a suitable route for high throughput in situ structural biology.

## Methods
### Dual beam plasma FIB/SEM microscopes
Milled samples and plasma characteristics data were acquired on either (i) a dual-beam focused ion beam scanning electron (FIB/SEM) "Helios Hydra" microscope (Thermo Fisher Scientific, Oregon, USA) equipped with a cryogenic stage and plasma multi-ion source (argon,

nitrogen, xenon, and oxygen) or (ii) a dual-beam focused ion beam scanning electron microscope with redesigned sample chamber and loading mechanism, which is a prototype for the commercially available Arctis microscope (Thermo Fisher Scientific, Oregon, USA). Briefly, this microscope was equipped with several modifications to enable plasma FIB milling within a small, enclosed, (ultra) high vacuum chamber.

Chamber/Autoloader/Stage: The chamber volume of the system is much smaller than the Helios Hydra, with a reduction in volume from ~0.04 m³ to <0.01 m³ (less than a 5th of the volume of a conventional Helios-type chamber). This enables a vacuum on the order of $1 \times 10^{-7}$ mbar to be achieved. It includes ports for the PFIB and SEM columns, cryogenic stage, robotic multi-specimen entry/exit (Autoloader), charge dissipation via a platinum metal sputter target and organo-metallic platinum protective coating chemistry via a gas injection system (GIS). Its volume reduces the contamination rates of a conventional chamber to <2 nm/hr. The chamber also comprises of a stage which allows loading of grids assembled into compatible mounts (autogrids) via the Autoloader. The autogrids are also compatible with the Titan Krios for imaging via TEM and can be shuttled between the instruments via a cassette without need for manual handling. Up to 12 samples may be stored and recalled from the autoloader to the main stage and then subsequently transferred to the TEM. The stage is enclosed with shielding which enables a clean working environment within the instrument needed to support long automation runs. The system is cooled via an Autofill system enabling long run times and unattended operation over days.

Electron Column: The electron optical column is a NiCol column (Thermo Fisher). The SEM comprises of a field emission gun assembly with Schottky-emitter, with a dual objective with field-free magnetic. The electron column provides detection in-lens using back scatter and secondary electrons. The detector is a T1/T2.

Plasma ion column: The PFIB column can switch between three ion species (xenon, argon, and oxygen) switchable plasma ion source. These species have both chemical and physical differences which allow for a range of milling effects.

Sputter Target: An ion sputtering device comprising a platinum mass is used to create thin layers of conductive platinum. Here, the primary ion source is used at high current to remove atoms of metal from a target allowing a dense vapour to flow over the sample. This renders samples conductive for imaging via SEM.

Gas injection system: Trimethyl(methylcyclopentadienyl)platinum(IV) (i.e., organo-platinum) is an organometallic compound which is used as a protective cap to protect the leading edge of the sample during milling. It enables lamella of sensitive samples to be protected by masking stray ions from affecting the delicate samples. This is typically condensed on the surface of the sample around a few micrometres thick as part of the preparation flow. The compound is evolved from a crucible contained within a delivery mechanism which generates vapour and delivers it via a needle to the chamber in small aliquots. The low temperature of the sample traps the vapour on contact, which aggregates to form the layer.

### Milling rate measurements
The milling rate was determined for each aperture commonly used during lamella preparation and on each beam by milling cross-sections with a controlled dose. Rates were determined at 30 kV.

All milling rates were determined on Thermo Scientific Helios Hydra with gas-cooled cryogenic stage. Perpendicular milling rates were determined using plunge-frozen yeast, on the thickest regions of yeast clumps. A schematic of the method for milling rate determination is shown in Supplementary Fig. 3a. It is well established that FIB milling is sensitive to surface topography[19]. To reduce these effects the sample was first coated in GIS and platinum, as is the case for lamella preparation. A ramp was then milled into this to produce a smooth

sample surface. The stage was then rotated and tilted so that now smooth sample surface of the ramp is presented perpendicular to the PFIB column for cross-sectional milling. Each condition was repeated three times and an average taken. The milled depth was measured using the SEM. For perpendicular milling, the SEM imaging occurs at an angle relative to the trench (52°). The perpendicular depth was therefore calculated using the angle of imaging (i.e., tilt correction). The milled cuboid volume was calculated using this perpendicular depth and the known milling area, as shown in Eq. (1).

$$volume = depth \times area \tag{1}$$

For the angular dependence study, plunge-frozen HeLa cells were used. In this case the stage was tilted after milling the ramps to introduce the required milling angle. The milled volume was assumed to be a parallelepiped (side profile of this is shown in Supplementary Fig. 3e) with two rectangular and two square faces; the volume in this case is given in Eq. (2). The volume was calculated by first calculating the area of the parallelogram (shown in yellow in Supplementary Fig. 3e) milled into the sample. The equations governing the relationship between the distances measured using the SEM and the milled volume dimensions are shown in Supplementary Fig. 3e. Theta ($\theta$) is the angle between the SEM and PFIB column, in our case 52°, and alpha ($\alpha$) is the milling angle relative to the milled ramp surface. Distances $b_m$ and $z_m$ were measured with the SEM without stage movement and trigonometry applied to work out the distances $b$ and $z$. From $z$ the distance $h$ can be calculated and therefore area of the parallelogram subsequently. The volume was obtained by multiplying this calculated area by the width, $w$, of the initial milling square pattern.

$$volume = \frac{wb_m z_m \sin(\alpha)}{\sin(\alpha + \theta)\sin(\theta)} \tag{2}$$

The milling rate in $\mu m^3/s$ was calculated for each mill by dividing the calculated volume by the milling time. Milling rates in $\mu m^3/nC$ were calculated by also dividing by the beam current that was measured for the chosen aperture.

Each mill was a standard square pattern with the Thermo Fisher Scientific default silicon milling settings, which by default will set the pitch size and the dwell time. The pitch size was set as suggested as it is sample independent but milling time was changed.

Error bars shown on the plots in Supplementary Fig. 3 were calculated by taking the standard deviation of the 18 repeats (three repeats for six different times) at each beam current. Errors shown in Table 2 are the standard error derived using the least squares method from a linear line of best fit on the milling rate ($\mu m^3/s$) against beam current (nA) plots. The linear line of best fit was forced to pass through the origin. Error bars shown in Table 3 are the same as those on plots shown in Supplementary Fig. 3g and h and are the standard deviation.

## Cell culture

UltrAuFoil on gold 200 mesh R2/2 grids (Quantifoil) were subjected to micropatterning[44,45] using the Primo module from Alveole, mounted on a Leica DMi8 microscope, following the manufacturer procedure. Briefly, the grids were coated with polylysine (100 µg/ml, 30 min) followed by mPEG-SVA (100 mg/ml, 1 h) and PLPP gel (1 h) prior to exposure to UV (50 mJ/mm²) to create circular patterns of 40 µm diameter. Then the grids were profusely rinsed with PBS before incubation with fibrinogen couple to Alexa 633 (Thermo Fisher). Micropatterned grids where then stored in Hank's Balanced Salt Solution (HBSS) (Gibco).

HeLa cells (CCL-2, ATCC, Manassas, VA, USA) were grown in Dulbecco's Minimal Essential Media (DMEM) (Gibco) with high glucose and non-essential amino acids complemented with 10% FBS, 1% glutamine and gentamycin (25 µm/ml). Cells were seeded on

micropattern grids for 2 h before washing. Infection with *Chlamydia trachomatis* LGV02 were performed as previously described[46]. 24 h post infection cells were plunge-frozen using the Vitrobot (Thermo Fisher) offsetting the blotting pad to favour back blotting. Just prior to plunge-freezing, the cells media was replaced with the complete media with 10% glycerol (v/v). Vitrified grids were then clipped (Thermo Fisher) into autogrids (Thermo Fisher) and subsequently stored under liquid nitrogen.

Plunge-frozen *S. cerevisiae* (Yeast) samples were prepared as previously described[43].

## Automated lamella fabrication

Autogrid clipped TEM grids were loaded into a cassette and loaded into the PFIB's multispecimen cassette which was then subsequently loaded onto the stage using the robotic sample delivery device (termed Autoloader) (Thermo Fisher Scientific). The SEM was used to screen the grids to ensure suitability for lamella preparation (Fig. 1a). Once selected for milling, grids were coated with tri-methyl(methylcyclopentadienyl)platinum(IV) using the gas injection system (GIS) and then sputter coated with platinum metal using an inbuilt platinum mass which was targeted with the PFIB beam at 16 kV and 1.4 µA to liberate clusters for adjacent surface metal coating.

Lamella sites were then identified and loaded into the AutoTEM cryo software (Thermo Fisher Scientific) with the stage at eucentric height. The milling was then carried out using 30 kV argon automatically by the software. The milling procedure was as follows: eucentric height was refined at each site before milling stress relief cuts 5 µm each side of the intended lamella using a 2.0 nA ion beam. Three rough milling steps were then used to remove material above and below the intended lamella position: (i) at 2.0 nA, 0.74 nA and 0.2 nA. This left a lamella of ~700 nm thick. This thicker lamella was then "polished" using 60 pA and 20 pA down to a nominal software target thickness of 70-90 nm. As the software is intended originally to operate with gallium, a considerable offset exists between the target thickness and the final lamella thickness for plasma ion sources. A software target thickness of 70-90 nm equated to ~200 nm in initial empirical tests for automation. Rough milling was carried out first on all lamella sites before proceeding onto the final polishing step. Drift corrected milling was used to ensure accurate milling. Each pattern was the default Thermo Fisher Scientific 'Rectangle' with silicon milling settings.

## Cryo-electron tomography

Multi-specimen cassettes with grids were directly transferred from the plasma FIB/SEM to the transmission electron microscope via an Autoloader (Thermo Fisher Scientific). Tomography data were acquired with a Titan Krios (Thermo Fisher Scientific) transmission electron equipped with a Falcon 4 camera and a Selectris energy filter. Dose-symmetric tilt-series were collected using Tomo 5 (Thermo Fisher scientific) software at a nominal magnification of 64000 x magnification (corresponding to a calculated magnification of 81081 x) at a 1.85 Å pixel size in electron counting mode from 51° to −51°, corrected for the pre-tilt of the lamellae, with 3° increments, with a total dose of 175 e⁻/Å². 

## Sub volume averaging

Warp version 1.0.9[47] was used for gain- and motion correction using 10 frame groups per tilt, contrast transfer function (CTF) correction and creating tilt-series stacks. Tilt-series were aligned and reconstructed at a binning factor 8 using AreTomo 1.1.1[48] and tomograms were flipped using IMOD 4.11[49], followed by bandpass filtering using EMAN 2.91[50]. Ribosomes were automatically picked using crYOLO 1.8.3[51] by creating a training dataset from five representative tomograms, followed by prediction on 180 tomograms, identifying 70436 putative ribosomes. Particles were extracted at a binning factor 4 using Warp with a box

size of 64×64×64 pixels and imported into RELION 3.1[52]. An initial reference was generated by performing 3D refinement on a random subset of 1000 particles, followed by 3D refinement with all the particles. 3D classification was performed with a circular mask, identifying 18119 particles as ribosomes. Another round of 3D classification with a tighter ribosome-shaped mask was performed, yielding 8 classes with 80 S ribosomes totalling 16204 particles, one class of the 60 S ribosome with 1896 particles and one junk-class with 19 particles. The 16204 particles from 80 S ribosome classes were extracted with warp at binning 2 with a box size of 128 × 128 × 128 pixels. Particle poses were subjected to 3D refinement in RELION using a thresholded ribosome mask. Particles poses for 15628 particles (six tomograms could not be refined post-RELION in M due to programme errors) were imported into M version 1.0.9[36] for five rounds of refinement where sequentially the following additional parameters were solved for: (1) 3 × 3 image warp grid and particle poses (2) 4 × 4 image warp grid (3) 3 × 3 x 2 × 10 volume warp grid (4) same settings as round 3 (5) stage angles and per-particle defocus estimation. Structures determined were fitted using a human ribosome model (PDB: 4UG0[53]), and the pixel size adjusted empirically to fit the PDB model. The determined pixel size for the map was 1.9 Å. This value was used for all FSC calculations (0.143) and B-factor calculations.

To produce structures of ribosomes at different distances from the lamella milling surfaces, a custom Python script was written to calculate the distance of particles from XYZ coordinates within the output from M/RELION. Briefly, boundary models were created manually for each tomogram every ~100 YZ slices using IMOD[49] and converted into a text files for use in the script. The script interpolates the manually created boundary models, determines the nearest distance for each ribosome coordinate to the surface model, determines the distance of the top and bottom surface model in the centre of the tomogram and generates STAR format files with additional columns for the distance for every particle and the thickness of the lamella for each tilt-series. The column of values can then be used as threshold for generating STAR files in RELION3.1[52]. Particles were then parsed into new STAR files based on their nearest distance to the top or bottom boundary model for 0-15 nm, 15-30 nm, 30-45 nm and 45-60 nm. STAR files for the matched controls with the same number of particles were created by randomly selecting particles from the same tomograms with a distance to the milling surfaces greater than the upper limit. In case not enough ribosomes with these criteria were present, ribosomes with a distance to the surfaces above the upper limit not already present in the list were randomly selected (for >15 nm; 1/351 particles, >60 nm; 226/2951 particles). The STAR files were then used for new rounds of sub-volume averaging in RELION 3.1. The local resolution was then determined after import into M, without any further refinement.

B-factors were calculated using RELION based on Rosenthal and Henderson[37]. Briefly, reconstructions were determined for subset of the particles described above. The determined resolutions were then plotted as a function of the particle number and the 2 over slope of the linear fit calculated (Fig. 4d; Supplementary Fig. 11).

## Data analysis
To determine the distance where tilt-series were recorded to the platinum layer in the front of the lamellae, search images saved by Tomo 5 were stitched in FIJI[54] and the distance from each position to the front of the platinum layer along the direction of the PFIB beam was measured with the line measurement tool.

## Reporting summary
Further information on research design is available in the Nature Portfolio Reporting Summary linked to this article.

## Data availability
Sub volume averages generated in this study have been deposited in the Electron Microscopy Data Bank (EMDB) under accession codes: EMD-15636 (full reconstruction), EMD-16196 (0 to 15 nm from milling surfaces), EMD-16199 (>15 nm from milling surfaces matched control), EMD-16185 (15 to 30 nm from milling surfaces), EMD-16186 (>30 nm from milling surfaces matched control), EMD-16192 (30 to 45 nm from milling surfaces), EMD-16193 (>45 nm from milling surfaces matched control), EMD-16194 (45 to 60 nm from milling surfaces) and EMD-16195 (>60 nm from milling surfaces matched control). The raw microscope data (frames and associated metadata for 180 tilt-series) used for subvolume averaging are available on the EMPIAR data server under accession code EMPIAR-11306. Source data are provided with this paper.

## Code availability
The software used to determine the distances of ribosomes from the PFIB milling surfaces is freely available on GitHub: https://github.com/rosalindfranklininstitute/RiboDist.

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

## Acknowledgements

The authors would like to thank Lu Gan, Alex de Marco, and Sebastian Tacke for robust discussions around milling rates. This work was supported by the Wellcome Trust through the Electrifying Life Science project (220526/Z/20/Z to J.H.N.). The Rosalind Franklin Institute is funded by UK Research and Innovation through the Engineering and Physical Sciences Research Council (EPSRC).

## Author contributions

M.D. prepared samples for cryo-PFIB. C.B., M.D., and T.G. performed cryo-PFIB, and collected cryoET data. C.B. and M.G. performed sub-volume averaging. N.B.-y.Y. and M.C.D. implemented new computation tools to help with data analysis. J.M.M., and Z.P., managed the development of the prototype cryo-PFIB/SEM instrument, with input from M.D., J.H.N and M.G. C.B., T.G., and M.G. prepared figures. J.H.N and M.G. supervised the project. C.B. and M.G. wrote the manuscript. All authors reviewed the data and commented on the manuscript.

## Competing interests

J.M.M. and Z.P. are employees of Thermo Fisher Scientific, which manufactures and sells certain equipment that was used in this research. All other authors declare no competing interests.
