## [Peer Review File · Nature Communications]

REVIEWER COMMENTS

Reviewer #1 (Remarks to the Author):

The authors describe their technical development and result using plasma beams. Plasma sources have comparable probe sizes to gallium at low current but have smaller probe sizes than gallium at higher beam currents. They conclusively show that plasma beams pose several advantages compared to gallium focused ion beams, which are prone to ion implantation and changes to the surface chemistry. In addition, they describe their development of automated plasma FIB milling of lamellas. These developments result in the production of hundreds of tomograms with an acclaimed 80% success rate. Lastly, they show the quality of their lamellas on the example of sub-tomogram averaged ribosomes.

The manuscript is written very clearly, is timely, and describes an important technical development that is already commercially available.

However, the manuscript falls short on two important issues that are mentioned in the beginning as advantages of the plasma beams.

1) It does not show that plasma beams do not deposit material on the surface of the lamellas or even implant material there, thus changing the surface chemistry.

2) It does not show that dose damage does not occur at the edge of the lamella, or in case that there is, how much and how deep this damage actually is.

It would be nice, if the authors could address these points, otherwise I recommend the publication.

Reviewer #2 (Remarks to the Author):

The authors report on the use of a prototype microscope equipped with a plasma ion source designed to produce cryo lamellae of vitrified biological samples. The manuscript illustrates very well the advantages of the improved microscope design. It clearly shows the advantages of this new design, that leads to longer automated run times, automated sample loading, lower contamination build up. The described workflow is shown to produce quality lamellae at reasonably high speed. The authors completed the lamellae preparation with a tomography and subtomogram averaging, which resulted in impressive in situ structures of human ribosomes at high resolutions of 4.9-8.2 Å. This is a clear and impressive story, that is of interest for the general audience and especially those interested in cryo-EM and technology development. I recommend this paper for publication with minor comments listed below.

- The free exchange of terms "sputter rate" and "milling rate" is very confusing. The authors should use just one term consistently. In my view "milling rate" is the most intuitive.

- Could the authors provide a few examples of the "smoother" surfaces generated by argon plasma vs. xenon for example? Or some kind of more quantitative argument for using argon instead of xenon. Otherwise at the moment it reads like, higher milling rates can be achieved by xenon plasma, but instead the slower argon plasma is chosen. This seems from the first glance a bit counterintuitive.

- Could the authors also comment, whether they envision the xenon (or other non-argon) plasma with higher milling rate can be used in the future to increase the milling speed even more? Even a factor of 2-3 seems like an important one timewise. This comment could be included in discussion.

- Could the authors please comment on the reason, why the average thickness for PFIB generated lamellae was thicker than the typical normal lamellae? Was this intentionally set to have ~250 nm thickness on average? Or is there another reason? In other words, can one get thinner lamellae with PFIB if desired? This comment can be included somewhere in the paragraph on page 6 lines 151-171.
- Units (Distance from leading edge) are missing from axes X in Fig.2

Reviewer #3 (Remarks to the Author):

Berger et al. describe the application of inductively coupled plasma ion sources to prepare thin electron transparent lamellae through biological samples for subsequent analysis with cryo-electron tomography and subtomogram averaging. The authors carefully evaluate sputter rates of different plasma species (xenon, nitrogen, oxygen and argon) on vitrified biological specimen. The authors then decided to use argon, even though it has the lowest sputter rate, for their first approaches to prepare cryo-TEM lamellae through HeLa cells in an automated manner. From 34 initially selected regions, 17 lamellae (50% success rate) were suitable for tilt-series acquisition, which revealed fully vitreous cellular features. A subsequent subtomogram average of ribosomes reached a global resolution of 4.9Å, which demonstrates the possibility to achieve high-resolution information from the prepared lamellae. In particular the author's observation that ribosomes closer to the lamella edge contain less higher spatial frequency information provide an interesting insight into the damaging effects of the plasma stream.

The application of plasma-FIB to prepare cryoTEM lamella has great potential to assess so far inaccessible voluminous samples for high-resolution cryoET analysis. Furthermore, due to the faster sputter rates of certain plasma ion species, this technique can significantly increase the throughput in lamella preparation in classical laboratory model systems, which is currently the bottleneck in many projects. It is unfortunate that the manuscript in its current form reports limited benefits of plasma ion sources over Ga. The combination of low throughput and success rate do not speak for a technique which has a 'suitable route for high throughput in situ structural biology' (line 297-298) when Ga provides similar if not better results at a fraction of the cost. The major improvement over commercially available Ga FIB milling instruments is the lower contamination rate due to the smaller chamber and integration into the cryo-ET workflow using an autoloader cassette.

That being said, this is the first study (next to a bioRxiv manuscript from Alex de Marco's lab, which should be cited [<https://www.biorxiv.org/content/10.1101/457820v1>]) demonstrating the feasibility and applicability of this technique to biological samples, providing the framework for future applications of pFIBs in the preparation of cryoTEM lamellae and is therefore of interest and worth being published. There are some comments that we think the authors need to address though.

- This is the first study, which shows the application of pFIB for cryoTEM lamella preparation. However, the assessment of the success rate is based on one single milling session. The authors should at least repeat (triplicate would be even better) their milling approach to get more insightful numbers to demonstrate the robustness of this method. The reported success rate of 50% is rather low. Previous work with autoTEM using Ga FIB milling showed a success rate of 88% (Tacke et al., 2021). Is plasma milling less robust? What improvements would be needed to achieve a similar success rate?
- The authors should provide SEM grid overview images pre- and post-milling. Furthermore, only one single lamella with significant ice contamination is shown. Is this their best

example? Please provide a gallery of TEM overview images of prepared lamellae.

- One of the big advantages of pFIB is the faster sputter rate, which could increase throughput or might even allow the preparation of TEM lamellae through bulky and voluminous samples, as stated in the introduction of the manuscript (line 69-71). The here presented approach using argon resulted in a total milling time of 45 min (line 136). This is a long time to mill a single lamella that is 12 μm wide (table 3). In previously published works on automated Ga FIB milling, the rate of lamella generation has been shown to be quicker albeit with slightly smaller lamellae. Why is the rate of lamella generation slower than with Ga? Demonstrating the feasibility of using different ion species to significantly speed up lamella preparation would be a great benefit for the community.

Other minor comments:

- Line 95: The acronym PFIB is introduced without ever defining that it stands for plasma focused ion beam.
- Line 111: Authors switch between plunge-frozen and plunge frozen.
- Line 116: 'Silicon' should be lowercase.
- Line 125: Can the authors provide a reason for the influence of the incident angle on the sputtering rate? This might have an impact on projects using the "waffle" method reported by Kelley et al. (<https://doi.org/10.1038/s41467-022-29501-3>).
- Line 134-137: The authors never state what sample is used to benchmark their milling approach. Only in line 174 they first mention HeLa cells.
- Line 137, Table 3 and lines 164-167: The milling depths being used were very small compared to the actual length of the lamellae. This is especially the case during polishing. Is this a possible reason for the large range in lamella thickness from the front to the back, especially in long lamella. Why aren't larger milling depths used? Furthermore, a drift correction every 10 or 30 s was used. Is this needed on the CompuStage? This might be also a factor contributing to the prolonged milling times.
- Line 158-159: Please state what the lamella thickness was that you aimed for during milling also in the manuscript main text. While in the methods a targeted software thickness of 70 to 90 nm is mentioned, the observed average lamella thickness in tomograms was $\sim 250\text{nm}$. Deviation from the target is common (<https://doi.org/10.7554/eLife.52286>) but how do the authors explain the large deviation they observe during plasma milling?
- Line 205 – 207: Do you expect similar results for Ga-ions or do the authors believe this is specific to the plasma ions used?
- Line 305-306: Is the instrument being used the newly released Thermo Fisher Arctis cryo-plasma-FIB?
- Line 415: Authors mention the acronym PF for the first time without introducing it.

Reviewer #1 (Remarks to the Author):

The authors describe their technical development and result using plasma beams. Plasma sources have comparable probe sizes to gallium at low current but have smaller probe sizes than gallium at higher beam currents. They conclusively show that plasma beams pose several advantages compared to gallium focused ion beams, which are prone to ion implantation and changes to the surface chemistry. In addition, they describe their development of automated plasma FIB milling of lamellas. These developments result in the production of hundreds of tomograms with an acclaimed 80% success rate. Lastly, they show the quality of their lamellas on the example of sub-tomogram averaged ribosomes.

The manuscript is written very clearly, is timely, and describes an important technical development that is already commercially available.

However, the manuscript falls short on two important issues that are mentioned in the beginning as advantages of the plasma beams.

1) It does not show that plasma beams do not deposit material on the surface of the lamellas or even implant material there, thus changing the surface chemistry.

Implantation of gallium in hard material science samples is well known (e.g. [10.1038/s41598-020-66564-y](https://doi.org/10.1038/s41598-020-66564-y)), and thus certain to occur in life science. Alternatives to gallium are important to explore for biological specimens. Our study establishes plasma ion sources as a viable alternative to gallium in structural biology.

The artefacts cannot be visually identified in reconstructed tomograms for life science samples, and as discussed in the discussion (lines 318 to 329) experimental determination would require technologies that we cannot currently perform.

We respectfully suggest that performing experiments to quantify ion implantation belongs to the future and outside of the scope of this study.

2) It does not show that dose damage does not occur at the edge of the lamella, or in case that there is, how much and how deep this damage actually is.

We have extended our analysis of the FIB damage layer to determine its depth and to separate this effect from partially ablated ribosomes very close the edges ("Ion beams.. 45 nm"; lines 190 to 220). This was done by creating separate reconstructions and determining the B-factors in 15 nm thick bands away from the milling surfaces (e.g. 0-15 nm, 15-30 nm) and controls with the same number of particles, but further away from the FIB milling surfaces. To exclude effects from variable lamella thickness and quality, the controls are matched to be from the same tomograms (where possible). The extended analysis is included in Figure 4, Supplementary Figures 10, the new Supplementary Figure 11 and the new table 4.

As we note this is an indirect measurement of damage.

It would be nice, if the authors could address these points, otherwise I recommend the publication.

We thank the reviewer for their recommendation for publication.

Reviewer #2 (Remarks to the Author):

The authors report on the use of a prototype microscope equipped with a plasma ion source designed to produce cryo lamellae of vitrified biological samples. The manuscript illustrates very well the advantages of the improved microscope design. It clearly shows the advantages of this new design, that leads to longer automated run times, automated sample loading, lower contamination build up. The described workflow is shown to produce quality lamellae at reasonably high speed. The authors completed the lamellae preparation with a tomography and subtomogram averaging, which resulted in impressive in situ structures of human ribosomes at high resolutions of 4.9-8.2 Å. This is a clear and impressive story, that is of interest for the general audience and especially those interested in cryo-EM and technology development. I recommend this paper for publication with minor comments listed below.

We thank the reviewer for their recommendation for publication.

- The free exchange of terms "sputter rate" and "milling rate" is very confusing. The authors should use just one term consistently. In my view "milling rate" is the most intuitive.

Physical science tends to use sputter rate and we are a mixed background team. We apologise. We have changed the manuscript to use the phrase "milling rate" throughout.

- Could the authors provide a few examples of the "smoother" surfaces generated by argon plasma vs. xenon for example? Or some kind of more quantitative argument for using argon instead of xenon. Otherwise at the moment it reads like, higher milling rates can be achieved by xenon plasma, but instead the slower argon plasma is chosen. This seems from the first glance a bit counterintuitive.

We now refer to another manuscript (Dumoux et al. 2022, DOI: 10.1101/2022.09.21.508877, currently in revision at elife) where we quantify curtaining rates for argon and xenon at different currents in the context of cryo-plasma FIB/SEM volume imaging and find lower curtaining propensity for argon, in particular at higher currents (line 132).

- Could the authors also comment, whether they envision the xenon (or other non-argon) plasma with higher milling rate can be used in the future to increase the milling speed even more? Even a factor of 2-3 seems like an important one timewise. This comment could be included in discussion.

We have added a sentence (line 251) to the discussion to not give the (possibly) false impression that xenon would be incompatible with automated lamella fabrication of similar quality to argon: "Although we focus in this study on using argon for lamella preparation, it's likely that with further

optimisation of the automated milling it should be possible to obtain similar results with xenon, for which we found higher milling rates than argon.”

We added to the discussion (lines 251 to 260) on potential speed increase with xenon and how lamella fabrication rates could be further increased, including with argon, and that any such potential increases in lamella fabrication rates should be balanced against success rate and the overall quality of the lamella.

- Could the authors please comment on the reason, why the average thickness for PFIB generated lamellae was thicker than the typical normal lamellae? Was this intentionally set to have ~250 nm thickness on average? Or is there another reason? In other words, can one get thinner lamellae with PFIB if desired? This comment can be included somewhere in the paragraph on page 6 lines 151-171.

An average thickness of 250 nm is on the thick side compared to manually prepared lamella [Berger et al. 2021], but similar average lamella thickness has previously been reported for automated lamella preparation with gallium [Zachs et al. 2020]. To reflect this, we have modified our discussion on thickness to incorporate this information and to describe thickness in terms of contribution of the damage layer (“Although manual.. lamella production”; lines 246 to 260).

We have added a section to the methods that describe how we determined a target thickness (lines 485 to 488).

We also outline a hybrid protocol, automated coarse milling of multiple lamellae in an unsupervised mode followed by manual polishing. This would be followed by a focused experiment of manual polishing of the lamellae and could generate thinner lamella should the practitioner wish (line 272).

We added a discussion on how thinner (and faster) lamellae could possibly be achieved, including manual polishing: “Although manually ... course milling; 246 to 260”.

We also discuss now how our results on the FIB damage layer depth may impact the optimal lamella thickness for subtomogram averaging (“However, it.. be advantageous”; lines 265 to 271).

We removed a sentence from the previous version of the manuscript where we speculate on the quality of our data in relation to the lamella thickness, as we have now discussed this more closely with regard to previous automated lamella fabrication [Zachs et al. 2020]: “We can only speculate why such high-quality data were obtained from lamella that would be regarded as “thick” (i.e. in comparison to a gallium milled example of ~185 nm).”

Berger, C., Ravelli, R.B.G., López-Iglesias, C., Kudryashev, M., Diepold, A., Peters, P.J., 2021. Structure of the Yersinia injectisome in intracellular host cell phagosomes revealed by cryo FIB electron tomography. *J. Struct. Biol.* 213, 107701. <https://doi.org/10.1016/j.jsb.2021.107701>

Zachs, T., Schertel, A., Medeiros, J., Weiss, G.L., Hugener, J., Matos, J., Pilhofer, M., 2020. Fully automated, sequential focused ion beam milling for cryo-electron tomography. *Elife* 9, 1–14. <https://doi.org/10.7554/eLife.52286>

- Units (Distance from leading edge) are missing from axes X in Fig.2

We added the missing units for the X axes in Fig. 2.

Reviewer #3 (Remarks to the Author):

Berger et al. describe the application of inductively coupled plasma ion sources to prepare thin electron transparent lamellae through biological samples for subsequent analysis with cryo-electron tomography and subtomogram averaging. The authors carefully evaluate sputter rates of different plasma species (xenon, nitrogen, oxygen and argon) on vitrified biological specimen. The authors then decided to use argon, even though it has the lowest sputter rate, for their first approaches to prepare cryo-TEM lamellae through HeLa cells in an automated manner. From 34 initially selected regions, 17 lamellae (50% success rate) were suitable for tilt-series acquisition, which revealed fully vitreous cellular features. A subsequent subtomogram average of ribosomes reached a global resolution of 4.9Å, which demonstrates the possibility to achieve high-resolution information from the prepared lamellae. In particular the author's observation that ribosomes closer to the lamella edge contain less higher spatial frequency information provide an interesting insight into the damaging effects of the plasma stream.

The application of plasma-FIB to prepare cryoTEM lamella has great potential to assess so far inaccessible voluminous samples for high-resolution cryoET analysis. Furthermore, due to the faster sputter rates of certain plasma ion species, this technique can significantly increase the throughput in lamella preparation in classical laboratory model systems, which is currently the bottleneck in many projects. It is unfortunate that the manuscript in its current form reports limited benefits of plasma ion sources over Ga. The combination of low throughput and success rate do not speak for a technique which has a 'suitable route for high throughput in situ structural biology' (line 297-298) when Ga provides similar if not better results at a fraction of the cost. The major improvement over commercially available Ga FIB milling instruments is the lower contamination rate due to the smaller chamber and integration into the cryo-ET workflow using an autoloader cassette.

That being said, this is the first study (next to a bioRxiv manuscript from Alex de Marco's lab, which should be cited [<https://www.biorxiv.org/content/10.1101/457820v1>]) demonstrating the feasibility and applicability of this technique to biological samples, providing the framework for future applications of pFIBs in the preparation of cryoTEM lamellae and is therefore of interest and worth being published. There are some comments that we think the authors need to address though.

We thank the reviewer for critically evaluating the manuscript. We believe to have improved on the original manuscript to address all the listed concerns. As suggested, we have also added the citation to the preprint from Alex de Marco's lab in the introduction.

- This is the first study, which shows the application of pFIB for cryoTEM lamella preparation. However, the assessment of the success rate is based on one single milling session. The authors should at least repeat (triplicate would be even better) their milling

approach to get more insightful numbers to demonstrate the robustness of this method. The reported success rate of 50% is rather low. Previous work with autoTEM using Ga FIB milling showed a success rate of 88% (Tacke et al., 2021). Is plasma milling less robust? What improvements would be needed to achieve a similar success rate?

We have extended our analysis on the success rate with two additional independent datasets with argon with the same protocol as used for the original data used in the paper. These are now included in Fig. 1 and communicated in the results (To determine.. and thickness; lines 136 to 156).

The 88% success rate reported by Tacke *et al.* was evaluated with the FIB/SEM microscope and should be compared to the 85% success rate we report here (with the two additional datasets) at the post FIB/SEM stage.

- The authors should provide SEM grid overview images pre- and post-milling. Furthermore, only one single lamella with significant ice contamination is shown. Is this their best example? Please provide a gallery of TEM overview images of prepared lamellae.

We added a gallery of 4 FIB images and matching SEM and TEM lamella overviews (Supplementary Figure 5). SEM grid overviews pre- and post-milling were added as Supplementary Figure 4. These have both been referenced on lines 138 and 139).

- One of the big advantages of pFIB is the faster sputter rate, which could increase throughput or might even allow the preparation of TEM lamellae through bulky and voluminous samples, as stated in the introduction of the manuscript (line 69-71). The here presented approach using argon resulted in a total milling time of 45 min (line 136). This is a long time to mill a single lamella that is 12 μm wide (table 3). In previously published works on automated Ga FIB milling, the rate of lamella generation has been shown to be quicker albeit with slightly smaller lamellae. Why is the rate of lamella generation slower than with Ga? Demonstrating the feasibility of using different ion species to significantly speed up lamella preparation would be a great benefit for the community.

Published data on the sputter rate for gallium (90°) is around 7.7 $\mu\text{m}^3/\text{nC}$ (DOI: 10.1116/1.2902962). This compares to argon in the present study at 4.5 $\mu\text{m}^3/\text{nC}$. Therefore, argon will be slightly slower than gallium. However, the higher currents will be accessible to argon ion beams and not gallium due to the nature of the probes at higher currents. Therefore, argon will indeed be useful at ablating more material for bulky samples.

Our study focused on argon for reasons now made clear in the manuscript (“While xenon.. not critical”; lines 130 to 136). We added a section in the discussion on how the lamella fabrication process could be further improved for speed, including work with xenon plasma, on explain the potential speed benefits that could be further optimised (“Although we.. important consideration”; 260 to 268).

We have avoided discussion of cost benefit ratio of PFIB vs GaFIB. We restricted our comments to the potential of PFIB to lead to high resolution cryoET structure.

Other minor comments:

- Line 95: The acronym PFIB is introduced without ever defining that it stands for plasma focused ion beam.
- Line 111: Authors switch between plunge-frozen and plunge frozen.
- Line 116: 'Silicon' should be lowercase.

We corrected these points in the manuscript.

- Line 125: Can the authors provide a reason for the influence of the incident angle on the sputtering rate? This might have an impact on projects using the "waffle" method reported by Kelley at al. (<https://doi.org/10.1038/s41467-022-29501-3>).

The angular dependence of milling rate is well established [Sigmund]. There is still some debate about the physical basis of this, but the generally accepted theory is that shallower collision cascades create a greater density of displaced atoms that can potentially be sputtered (as they are closer to the surface), leading to an increase in milling rate at more grazing incident angles. However, approaching very grazing incident angles (ie. close to 0° milling angle) the number of ions penetrating the surface drops rapidly, as more are reflected from the surface. [Sigmund]. We have added a summary of this in the paper on lines 230 to 235.

Sigmund, P., 1969. Theory of Sputtering. I. Sputtering Yield of Amorphous and Polycrystalline Targets, Phys. Rev. 184, 2, 383-416, <https://doi.org/10.1103/PhysRev.184.383>

- Line 134-137: The authors never state what sample is used to benchmark their milling approach. Only in line 174 they first mention HeLa cells.

We now mention the sample in the first paragraph on benchmarking the lamella milling approach.

- Line 137, Table 3 and lines 164-167: The milling depths being used were very small compared to the actual length of the lamellae. This is especially the case during polishing. Is this a possible reason for the large range in lamella thickness from the front to the back, especially in long lamella. Why aren't larger milling depths used? Furthermore, a drift correction every 10 or 30 s was used. Is this needed on the CompuStage? This might be also a factor contributing to the prolonged milling times.

The values for the milling depth in AutoTEM are derived from silicon, not HeLa cells. We iteratively improved the milling depths used in the different milling steps on biological sample to balance speed, success rate, curtaining rate and lamella thickness, before collecting the datasets presented in this manuscript.

The gradient in lamella thickness is a well-known phenomena caused by the gaussian profile of the FIB beam [Schaffer. *et al.* 2017].

We did not observe large drifts with this stage. We agree that less frequent drift correction may speed up milling times to some extent and could be further optimised in future experiments.

Schaffer, M., Mahamid, J., Engel, B.D., Laugks, T., Baumeister, W., Plitzko, J.M., 2017. Optimized cryo-focused ion beam sample preparation aimed at in situ structural studies of membrane proteins. *J. Struct. Biol.* 197, 73–82. <https://doi.org/10.1016/j.jsb.2016.07.010>

- Line 158-159: Please state what the lamella thickness was that you aimed for during milling also in the manuscript main text. While in the methods a targeted software thickness of 70 to 90 nm is mentioned, the observed average lamella thickness in tomograms was ~250nm. Deviation from the target is common (<https://doi.org/10.7554/eLife.52286>) but how do the authors explain the large deviation they observe during plasma milling?

In the results section we now state on line 151: The target thickness determined empirically was approx. 200 nm (see methods for details).” The methods then specify on line 487 to 488 “A software target thickness of 70 – 90 nm equated to approximately 200 nm in initial empirical tests for automation”.

This was done because the target thickness parameter used by AutoTEM software is currently not yet optimized for plasma ion species. This is described in the materials and methods section on lines 485 to 487: “As the software is intend originally to operate with gallium, a considerable offset exists between the target thickness and the final lamella thickness for plasma ion sources”.

- Line 205 – 207: Do you expect similar results for Ga-ions or do the authors believe this is specific to the plasma ions used?

Based on published results of both empirical and simulated data in material science (Burnett *et al.* 2016, Liu *et al.* 2020), gallium ions are likely to create a deeper damage layer on vitreous biological samples compared to xenon. We added this near the end of the discussion (lines 318 to 329):

“Based on findings in material science applications^{33,34}, gallium is also likely to create a damage layer near the milling surfaces for vitreous biological samples, quite possibly with a greater depth.”

Burnett, T.L., Kelley, R., Winiarski, B., Contreras, L., Daly, M., Gholinia, A., Burke, M.G., Withers, P.J., 2016. Large volume serial section tomography by Xe Plasma FIB dual beam microscopy. *Ultramicroscopy* 161, 119–129. <https://doi.org/10.1016/j.ultramic.2015.11.001>

Liu, J., Niu, R., Gu, J., Cabral, M., Song, M., Liao, X., 2020. Effect of Ion Irradiation Introduced by Focused Ion-Beam Milling on the Mechanical Behaviour of Sub-Micron-Sized Samples. *Sci. Rep.* 10, 10324. <https://doi.org/10.1038/s41598-020-66564-y>

- Line 305-306: Is the instrument being used the newly released Thermo Fisher Arctis cryo-plasma-FIB?

The FIB/SEM microscope used for all the lamella preparation was a custom prototype for the Arctis. We have added this in the materials and methods section (line 350 to 351) where we describe the the FIB/SEM microscopes: “.., which is a prototype for the commercially available Arctis microscope

- Line 415: Authors mention the acronym PF for the first time without introducing it.

We replaced the two occurrences of the acronym with plunge-frozen or plunge-freezing.

REVIEWERS' COMMENTS

Reviewer #3 (Remarks to the Author):

The authors made an effort in improving the manuscript and addressed the points from the first review clearly.

A couple of minor points are listed below:

Fig 1. and Suppl. Fig. 5:

It took us a long time to understand the origin of the electron-dense bar on every milled lamella, but we concluded that this is probably the UltrAuFoil of the used grid type. As this might be a potential cause for misunderstandings, a small comment on this in the figure legend would be helpful. On top, it would be helpful to include a statement about why this grid type was used and if the authors see a benefit of using UltrAuFoil for pFIB milling approaches.

Line 217: the comma is not in the right place it should be "...milling surfaces, put to a..."

Sup. Fig. 6: missing the labels for the y-axis

Sup. Fig. 6: b/c link to Figure 4d is not clear to us. line 857: "(middle and right dataset in Figure 4d for panel b and c in this figure respectively)"

Line: 460: It is not mentioned anywhere in the manuscript, that HeLa cells infected with Chlamydia were used in this study?

Reviewer #3:

Fig 1. and Suppl. Fig. 5:

It took us a long time to understand the origin of the electron-dense bar on every milled lamella, but we concluded that this is probably the UltrAuFoil of the used grid type. As this might be a potential cause for misunderstandings, a small comment on this in the figure legend would be helpful. On top, it would be helpful to include a statement about why this grid type was used and if the authors see a benefit of using UltrAuFoil for pFIB milling approaches.

We specified that we used UltrAuFoil in the caption of figure 2 and supplementary figure 1 and supplementary figure 5, and added in the caption of supplementary figure 5 that the gold of the foil of the grid is visible as an electron-dense band on every lamella.

Line 217: the comma is not in the right place it should be "...milling surfaces, put to a..."

We corrected the incorrectly placed spaces

Sup. Fig. 6: missing the labels for the y-axis

We added a label for the y-axes in supplementary figure 6.

Sup. Fig. 6: b/c link to Figure 4d is not clear to us. line 857: "(middle and right dataset in Figure 4d for panel b and c in this figure respectively)"

We removed the unclear description and instead added the number of lamellae the tilt-series were acquired from in the caption of Figure 2d, so that the three datasets in Figure 1, 2d and Sup. Fig. 6b/c can be matched by the reader.

Line: 460: It is not mentioned anywhere in the manuscript, that HeLa cells infected with Chlamydia were used in this study?

We added in the main text of the results section that cells were infected with chlamydia: (chlamydia infected) HeLa cells.